Pattern analysis of total item score and item response of the Kessler Screening Scale for Psychological Distress (K6) in a nationally representative sample of US adults

Tomitaka Shinichiro tomitaka.shinichiro@jp.panasonic.com 1 2
Kawasaki Yohei 2 3
Ide Kazuki 2 3 4
Akutagawa Maiko 2
Yamada Hiroshi 2
Yutaka Ono 5
Furukawa Toshiaki A. 6
1 Department of Mental Health, Panasonic Health Center , Japan
2 Department of Drug Evaluation and Informatics, School of Pharmaceutical Sciences, University of Shizuoka , Shizuoka , Japan
3 Department of Pharmacoepidemiology, Graduate School of Medicine and Public Health, Kyoto University , Kyoto , Japan
4 Center for the Promotion of Interdisciplinary Education and Research, Kyoto University , Kyoto , Japan
5 Center for the Development of Cognitive Behavior Therapy Training , Tokyo , Japan
6 Department of Health Promotion and Human Behavior, Department of Clinical Epidemiology, Kyoto University Graduate School of Medicine, School of Public Health, Kyoto University , Kyoto , Japan
Leach Liana
Electronic publication date: 2017 Feb 9
Publication date: 2017
Volume: 5
Electronic Location ID: e2987
Received 2016 Oct 21; Accepted 2017 Jan 12
Copyright: ©2017 Tomitaka et al.
Copyright year: 2017
Copyright holder: Tomitaka et al.
License: This is an open access article distributed under the terms of the Creative Commons Attribution License, which permits unrestricted use, distribution, reproduction and adaptation in any medium and for any purpose provided that it is properly attributed. For attribution, the original author(s), title, publication source (PeerJ) and either DOI or URL of the article must be cited.
License URL: https://creativecommons.org/licenses/by/4.0/

Keywords: Depressive symptoms, K6, Kessler screening scale, Latent trait, Exponential distribution, Item response theory, Likert scale, General population, MIDUS, CES-D

Funding: The authors received no funding for this work.

==============================
Background

Several recent studies have shown that total scores on depressive symptom measures in a general population approximate an exponential pattern except for the lower end of the distribution. Furthermore, we confirmed that the exponential pattern is present for the individual item responses on the Center for Epidemiologic Studies Depression Scale (CES-D). To confirm the reproducibility of such findings, we investigated the total score distribution and item responses of the Kessler Screening Scale for Psychological Distress (K6) in a nationally representative study.

Methods

Data were drawn from the National Survey of Midlife Development in the United States (MIDUS), which comprises four subsamples: (1) a national random digit dialing (RDD) sample, (2) oversamples from five metropolitan areas, (3) siblings of individuals from the RDD sample, and (4) a national RDD sample of twin pairs. K6 items are scored using a 5-point scale: “none of the time,” “a little of the time,” “some of the time,” “most of the time,” and “all of the time.” The pattern of total score distribution and item responses were analyzed using graphical analysis and exponential regression model.

Results

The total score distributions of the four subsamples exhibited an exponential pattern with similar rate parameters. The item responses of the K6 approximated a linear pattern from “a little of the time” to “all of the time” on log-normal scales, while “none of the time” response was not related to this exponential pattern.

Discussion

The total score distribution and item responses of the K6 showed exponential patterns, consistent with other depressive symptom scales.

Introduction

Depression is a major public health concern and is one of the leading causes of disease burden worldwide (Moussavi et al., 2007). Because depressive symptomology (number and chronicity) form the basis of a diagnosis of clinical depression, there is considerable interest in understanding the distribution of depressive symptoms in the general population (Blazer & Kessler, 1994; Kroenke et al., 2009). However, even though representative values of the distributions of item scores, including an average value and a median value, has been overwhelmingly investigated by many researchers, the mathematical pattern of the total score distribution and item responses on measures of depressive symptoms in a general population remains unknown.

To this end, several recent studies with large sample sizes determined that total scores on measures of depressive symptoms in the general population approximate an exponential distribution, except at the lowest end of the range of scores. As shown in Fig. 1, the graph of an exponential distribution starts on the y-axis at a positive value (λ) and decreases to the right at a constant rate. The graph of an exponential distribution follows a linear pattern with a log-normal scale.

In an analysis of data on nearly 10,000 non-psychotic respondents of the British National Household Psychiatric Morbidity Survey, Melzer et al. (2002) determined that an exponential curve provided the best fit to total depressive and neurotic symptom scores on the Revised Clinical Interview Schedule (CIS-R). Furthermore, using data from nearly 32,000 respondents of a national survey of the Japanese population (Ministry of Health, Labor and Welfare, Statistics and Information Department, 2002; Tomitaka, Kawasaki & Furukawa, 2015a), we similarly observed that the distribution of total scores on the Center for Epidemiologic Studies Depression Scale (CES-D) follows an exponential curve, except at the lowest end of the scale. These results were repeated in a large sample of nearly 8,000 Japanese employees (Tomitaka et al., 2016a).

Furthermore, we recently determined that the exponential pattern is present for the individual item responses on the CES-D (Tomitaka, Kawasaki & Furukawa, 2015b). The CES-D allows an individual to self-rate the frequency of a variety of depressive symptoms during the past week with response options of “rarely (less than 1 day),” “some of the time (1 to 2 days),” “occasionally (3 to 4 days),” and “most of the time (5 to 7 days)”; (Radloff, 1977). In an analysis of the data from the same nationally representative survey as for the total score analysis, we demonstrated that the distributions of the 16 negative items on the CES-D commonly exhibited exponential patterns between the “some of the time” and “most of the time” responses, while the “rarely” response was not related to this exponential pattern (Tomitaka, Kawasaki & Furukawa, 2015b).

Figure 1 Distribution of an exponential distribution.

If a random variable, x is exponentially distributed, f(x) = λe−λx for x ≥ 0 where λ is the rate parameter. The graph of an exponential distribution starts on the y-axis at a positive value (λ) and decreases to the right.

Taken together, these findings suggest that the depressive symptom items are manifest variables of the unidimensional latent trait of depressive symptoms, which follows an exponential distribution (Tomitaka, Kawasaki & Furukawa, 2015b; Tomitaka et al., 2016b). According to this notion, regardless of the number or choice of specific depressive symptom items, total scores and item responses will exhibit the same characteristic patterns as observed in the CES-D and CIS-R. However, there is no evidence of whether these findings would be reproduced for other depressive symptom scales.

The Kessler Screening Scale for Psychological Distress (K6) is a widely used screening scale for mental health problems (Kessler et al., 2002). It measures the severity of psychological distress (depressed mood, motor agitation, fatigue, worthless guilt and anxiety) and identifies people who may have a high likelihood of a diagnosable mental illness (Furukawa et al., 2003). The K6 has been included as part of the National Survey of Midlife Development in the United States (MIDUS) since 1995 and the National Health Interview Survey (NHIS) since 1997. Furthermore, the K6, or the slightly more comprehensive K10, has been used in general population surveys in Australia, Canada, Japan, and the US, and as part of the surveys of the World Health Organization in 30 other countries worldwide (Kessler & Üstün, 2004). It would be clinically important to understand the patterns of total score distribution and item responses of the K6 scale.

Furthermore, the K6 was developed using item response theory, which generally assumes that a normally distributed latent trait underlies the performance of a manifest variable (Kessler et al., 2002; Embretson & Reise, 2000). However, to the best of our knowledge, there is little evidence that depressive symptoms follow a normally distributed latent trait. Although estimated parameters can be obtained using item response theory or factor analysis, these statistical procedures presuppose a specific distribution type for the latent trait, implying that it is distinct from estimating the type of the latent trait distribution on ordinal scale variables (Tennant & Conaghan, 2007; Gollini & Murphy, 2014). We raise the possibility that the latent trait of depressive symptoms is an exponential distribution. The hypothesis is supported by the findings that the total score distribution and item responses follow exponential patterns. Thus, to investigate the distribution type of the latent trait of depressive symptoms, it would be useful to analyze the pattern of total score distribution and item responses for the K6 items.

For the present study, we used some of the data from the first wave of the MIDUS (i.e., the MIDUS 1) (ICPSR 2760, 2016). It is freely available to researchers and has been used in hundreds of papers in psychology, sociology, and public health (National Survey of Midlife Development in the US (Midus), 2016). MIDUS 1 dataset comprises individuals from four subsamples: (1) a national random digit dialing (RDD) sample, (2) oversamples from five metropolitan areas in the US, (3) siblings of individuals from the RDD sample, and (4) a national RDD sample of twin pairs. By analyzing all four subsamples of the MIDUS 1, we determined whether the K6 also exhibits the same patterns of total score distribution and item responses observed in previous studies on the CES-D and CIS-R.

Materials and Methods

Dataset and analysis procedure

Data were drawn from the MIDUS 1 (ICPSR 2760, 2016). The MIDUS 1 is a nationally representative sample of non-institutionalized, English-speaking adults aged 25–74 from the US. Conducted by the MacArthur Foundation Research Network on Successful Midlife Development from 1994 to 1995, the MIDUS 1 comprises a total of 7,108 participants (3,395 men) of an average age of 46.4 years old (SD = 13 years). As noted previously, the MIDUS 1 comprised individuals from four subsamples: (1) a national RDD sample (n = 3,487), (2) oversamples from five metropolitan areas in the US (n = 757), (3) siblings of individuals from the RDD sample (n = 950), and (4) a national RDD sample of twin pairs (n = 1,914).

The MIDUS 1 comprised a telephone interview of approximately 30 min and two self-administered questionnaires, each of which was approximately 45 pages in length. All participants provided informed consent, and the MIDUS was approved by the institutional review board at each data collection site.

The main RDD sample was selected and invited to participate using telephone banks. For each household contacted, a list was generated of all people in that household aged between 25 and 74 years old, from which a respondent was randomly selected. The MIDUS 1 researchers also carried out oversampling of five metropolitan areas. Of the RDD respondents who reported having one or more siblings, 529 were randomly selected and complete telephone interviews were conducted with 950 of their siblings, some of whom came from the same household. Finally, twin pairs were recruited from a representative national sample of approximately 50,000 households for the presence of a twin. The response rates and select sociodemographic characteristics of the MIDUS 1 samples are reported in detail elsewhere (ICPSR 2760, 2016; Brim, Ryff & Kessler, 2004).

Ethics statement

The present paper is a secondary analysis of freely available data so format clearance was not needed. As our local institutional review board does not consider de-identified secondary data analysis to be human subjects research, our study did not require ethical approval from that board.

Measures and analysis

In the self-administered questionnaire of the MIDUS 1, depressive symptoms were assessed using the K6. The K6 comprises six items that ask about the frequency that participants felt sad, nervous, restless, hopeless, that everything was an effort, and worthless during the past 30 days. Each item is rated on a 5-point scale with options of 0 = none of the time, 1 = a little of the time, 2 = some of the time, 3 = most of the time, and 4 = all of the time, yielding a total item score of 0–24. Higher scores indicate greater psychological distress. One of the K6 items used in the MIDUS and NHIS surveys is worded as follows: “How much of the time did you feel so sad nothing could cheer you up.” This contrasts with the wording generally used in the K6 today: “How often did you feel so depressed that nothing could cheer you up?” (Kessler et al., 2002).

Participants who did not respond to all K6 items were excluded from the total score analysis. The final sample for the total score analysis comprised 2,975, 647, 858, and 1,743 individuals from the national RDD sample, oversamples, siblings, and twin pairs, respectively. To evaluate whether K6 total scores follow an exponential pattern, the distributions of the total score were analyzed using a log-normal scale and an exponential regression model.

For the item analysis, participants who did not respond to each item were excluded from the item response analysis (Table 1 and Table S1). We analyzed the item responses for the six items using normal and log-normal scales. The regression curve for the exponential model was estimated using the least squares method. JMP Version 11 for Windows (SAS Institute, Inc., Cary, NC, USA) was used to calculate descriptive statistics and the frequency distribution curves.

Table 1 Item responses of a national RDD subsample.

A similar pattern was observed in response rates for all 6 items, with the highest response frequency being for “none” and a decreasing frequency thereafter as item scores increased, and the lowest response frequency observed for “all of the time.” There were no exceptions to this pattern.

Item	Response number (%)	
	None	A little	Some	Most	All	
Sad	2,058 (68.3)	637 (21.1)	250 (8.3)	54 (1.8)	13 (0.4)	
Nervous	1,297 (43.0)	1,079 (35.8)	516 (17.1)	97 (3.2)	24 (0.8)	
Restless	1,375 (45.7)	1,011 (33.6)	502 (16.7)	84 (2.8)	35 (1.2)	
Hopeless	2,372 (78.7)	391 (13.0)	173 (5.8)	51 (1.7)	20 (0.7)	
Effort	1,684 (56.1)	802 (26.7)	342 (11.4)	129 (4.3)	46 (1.5)	
Worthless	2,372 (78.7)	391 (13.0)	184 (6.1)	42 (1.4)	25 (0.8)	
average	1,860 (61.8)	719 (23.9)	328 (10.9)	76 (2.5)	27 (0.9)	

Results

K6 total score analysis

Figure 2 depicts the distribution of K6 total scores for (A) the national RDD sample, (B) oversamples from five metropolitan areas, (C) the siblings of the RDD sample, and (D) the national RDD sample of twin pairs. While the total score distributions for the four subsamples were commonly right-skewed, the frequencies of the zero score were different across groups due to the difference in sample sizes.

Figure 2 Distributions of K6 total scores.

(A) National RDD sample (n = 2,975) , (B) national RDD sample of twin pairs (n = 1,743), (C) siblings of individuals from the RDD sample (n = 858), and (D) oversamples from five metropolitan areas in the US (n = 647). While the distributions of the K6 total scores for the four groups are commonly right-skewed, the frequencies of the zero score differed across groups.

All the four sub-samples showed linear patterns with similar gradient on the log-normal scale, suggesting that the K6 total scores followed an exponential pattern with similar rate parameters (Fig. 3). While the national RDD subsample and the twin pairs exhibited a linear pattern for almost the entire range of total score (Figs. 3A and 3B), the frequencies of the sibling and the oversample subsamples were often zero over scores of 16–18 because of the small sample sizes (Figs. 3C and 3D). In fact, it is likely that the small sample sizes for the higher scores governed the increasing fluctuation in lines for each subsample as total score increased.

Figure 3 Distributions of K6 total scores on log-normal scales.

(A) National RDD sample, (B) national RDD sample of twin pairs, (C) siblings of individuals from the RDD sample, and (D) oversamples from five metropolitan areas in the US. All four subsamples showed linear patterns with similar gradients.

The regression curves for the exponential model were calculated for the national RDD sample (y = 860.9e−0.259x, R2 = 0.95), oversamples from five metropolitan areas (y = 189.46e−0.264x, R2 = 0.96), siblings of individuals from the RDD sample (y = 249.9e−0.274x, R2 = 0.95), and national RDD sample of twin pairs (y = 470.8e−0.261x, R2 = 0.94). The independent variable, x is the K6 total score and the dependent variable, y is the number of subjects. R2 is the coefficient of determination. This analysis revealed high coefficients of determination with similar parameters in all four subsamples, thus indicating that the total K6 scores fit well to an exponential distribution with similar rate parameters.

Item response analysis

Table 1 shows the item response rates for each of the 6 items in the national RDD subsample. A similar pattern was observed in response rates for all 6 items, with the highest response frequency being for “none” and a decreasing frequency thereafter as item scores increased, and the lowest response frequency observed for “all of the time.” There were no exceptions to this pattern.

To evaluate the pattern of item responses, all 6 item response rates were plotted together on the same graph. As shown in Fig. 4A, the item responses of the six items showed a common pattern, which displays different types of distributions with a boundary at “a little of the time.” As indicated by the arrow in Fig. 4A, the lines for the six items crossed at a single point between “none of the time” and “a little of the time,” after which they showed a decreasing pattern. As verified in our previous study, if the response distributions of the 6 items follow an exponential pattern with the similar parameters between “a little of the time” and “all of the time,” then all of the lines will inevitably cross at a single point between “none of the time” and “a little of the time” (Tomitaka, Kawasaki & Furukawa, 2015b).

Figure 4 The item responses of K6.

(A) National RDD sample, (B) national RDD sample of twin pairs, (C) siblings of individuals from the RDD sample, and (D) oversamples from five metropolitan areas in the US. All four subsamples showed that the item response for each of the six items exhibited a common pattern with a boundary between “none of the time” and “a little of the time.” As indicated by the arrow (Fig. 3A), the lines for the six items crossed at a single point between “none of the time” and “a little of the time,” whereas the lines from “a little of the time” to “all of the time” showed a pattern of reduction.

The item response rates for the remaining three subsamples exhibited a similar pattern as for the national RDD subsample—the highest response frequency was for “none of the time,” after which the response frequency decreased with increasing item score (such that the lowest response frequency was observed for “all of the time”; Table S1). No exceptions to this pattern were observed. Furthermore, as depicted in Figs. 4B– 4D, the lines for the 6 items crossed at a single point between “none of the time” and “a little of the time,” after which they showed a decreasing pattern.

On the log-normal scale, the item response rates approximated a linear pattern for the “a little of the time” to “all of the time” responses across all four subsamples (Fig. 5). In addition, the gradients of the linear patterns for the 6 items were rather similar. Specifically, the lines for “a little of the time” to “some of the time” were somewhat parallel, whereas the lines for “most of the time” to “all of the time” were less so.

Figure 5 The item responses of K6 on log-normal scales.

(A) National RDD sample, (B) national RDD sample of twin pairs, (C) siblings of individuals from the RDD sample, and (D) oversamples from five metropolitan areas in the US. Using a log-normal scale, all four subsamples showed that the item responses for the 6 items followed a linear pattern between “a little of the time” and “all of the time.” In addition, the gradients of the linear patterns for the six items were similar to each other. To be specific, the lines for “a little of the time” to “some of the time” were relatively parallel to each other, whereas the lines for “most of the time” to “all of the time” were less so.

Discussion

Our aim was to determine whether the total score distribution and item responses of the K6 exhibit the exponential pattern observed in previous studies on the CES-D and CIS-R. The main findings are as follows: (1) regardless of subsample, the K6 total scores approximate an exponential pattern with similar rate parameters, and (2) the K6 item responses exhibit exponential patterns between the “a little of the time” and “all of the time” responses, while the “rarely” response is not related to this exponential pattern.

K6 total scores approximated an exponential pattern

Our findings indicate that K6 total scores of a representative sample of US adults approximate an exponential pattern, which is consistent with the results of other national representative surveys in England and Japan using the CIS-R and CES-D, respectively (Melzer et al., 2002; Bebbington et al., 2013; Tomitaka, Kawasaki & Furukawa, 2015a). Although the K6, CIS-R, and CES-D differ in terms of the number of items, item content, and scoring methods (the CIS-R uses a binary response scale, while the CES-D uses a four-point scale), the exponential pattern of total scores in a representative population was confirmed within all three scales. Taken together, the current evidence suggests that the total score of depressive symptom measures approximates an exponential pattern irrespective of the choice of items or the scoring method. In support of this conclusion, the previous study using the CES-D has demonstrated that, for any number or combination of items, the sum of the item scores of depressive symptoms approximates an exponential pattern (Tomitaka et al., 2016d). Of course, the depressive symptom items mentioned here do not include the reverse-scored positive affect items because our previous studies using the CES-D showed that reverse-scored positive affect items were not considered manifest variables of the latent trait of depressive symptoms (Tomitaka, Kawasaki & Furukawa, 2015b; Tomitaka et al., 2016c).

The reason that the total scores for the depressive symptoms scale approximates an exponential pattern might be explained by a theory that our research group has proposed (Tomitaka, Kawasaki & Furukawa, 2015a; Tomitaka et al., 2016b; Tomitaka et al., 2016c). The theory comprises the following three conditions: (1) depressive symptom items are manifest variables influenced by a unidimensional latent trait of depressive symptoms; (2) the latent trait of depressive symptom items follow an exponential distribution; and (3) the threshold of ordinal scale scoring forms a distribution in accordance with the unidimensional latent trait. The three conditions were devised based on earlier findings concerning how the total score distribution, the item responses and boundary curves of the total scores of depressive symptoms measures exhibit an exponential pattern (Tomitaka, Kawasaki & Furukawa, 2015b; Tomitaka et al., 2016c). In support of this theory, our previous study simulating these three conditions confirmed that the total score of depressive symptom items approximates the exponential pattern of a latent trait distribution, except for the lower end of the distribution (Tomitaka et al., 2016b).

Considering the rate parameters of the exponential models of total score distribution, the estimated parameters were similar across the four subsamples (−0.26 to −0.27). Our previous analysis of a national representative survey using the CES-D demonstrated that the rate parameters of the exponential model of summed scores were similar across the groups with the same number of items and the rate parameters of the summed item scores increased as the number of summed items increased (Tomitaka et al., 2016d). Interestingly, the estimated rate parameters of the present four subsamples were similar to those of samples that used the sum of eight CES-D items (−0.26 to −0.29), not the sum of two (−0.62 to −0.93), four (−0.41 to −0.52) or 16 CES-D items (−0.14). The similarity of the estimated rate parameters between the K6 and the sum of the eight CES-D items might be attributed to the fact that both the K6 and the sum of the eight CES-D items had the same maximum total score of 24 points. According to the aforementioned theory, a maximum total score means the number of thresholds in accordance with the latent trait. Further mathematical explanation is necessary to elucidate the mechanism of the rate parameter variance.

Although several groups have pointed out that total depressive symptomatic scores do not follow an exponential curve at the lower end of the distribution (Melzer et al., 2002; Bebbington et al., 2013), the K6 total scores of the present studies exhibited an exponential pattern for almost the entire range of total score (Fig. 3). The previous studies demonstrated that the probability of “none of the time” is a key index to predict the non-exponential pattern at the lower end of the distributions (Tomitaka et al., 2016d). Specifically, the sum of the item scores with high probability of “none of the time” exhibited higher scores compared to those predicted from the exponential pattern, whereas the sum of the item scores with low probability of “none of the time” exhibited lower scores compared to those predicted at the lower end of the distributions. In fact, according to our estimation, the mean probability of “none of the time” in the present studies (61.8%) is close to that of the CES-D 16 items (67.7%), which exhibited an exponential pattern for almost the entire range of total score (Tomitaka et al., 2016d).

Item response of K6 items

Consistent with the results using the CES-D (Tomitaka, Kawasaki & Furukawa, 2015b), the item responses of the K6 for the four subsamples exhibited a common pattern with a boundary at “a little of the time.” Furthermore, the lines for the 6 items crossed at a single point between “none of the time” and “a little of the time,” while the item responses from “a little of the time” to “all of the time” approximated a linear pattern on log-normal scales.

The mechanism of this pattern of the K6 items might be speculated as follows (Tomitaka, Kawasaki & Furukawa, 2015b). First, each item of the K6 is rated in two stages. First, each subject determines whether a symptom is present; if the symptom’s severity does not exceed the threshold for everyday mild symptoms, then it is regarded as “absent” (i.e., it occurs “none of the time”). Second, if the depressive symptom does meet the threshold, then the duration of the symptom is quantified according to the item response format as “a little of the time,” “some of the time,” “most of the time” and “all of the time.” This two-step process increases the likelihood that “none of the time” will cover the range of symptoms that does not meet the threshold, while the remaining response options (“a little of the time,” “some of the time,” “most of the time” and “all of the time”) cover about one fourth of the above-threshold range. If the latent trait of depressive symptoms follows an exponential distribution, which uniquely has a memoryless property, the item responses for the K6 would all exhibit the mathematical distribution observed in the present study. Due to memoryless property, regardless of the difference of the threshold of each item, the decreasing ratio of the probabilities among “a little of the time,” “some of the time,” “most of the time” and “all of the time” is constant. However, as this remains a mere speculation, further research would be needed.

In the present study, the K6 items followed an exponential pattern with similar rate parameters between responses of “a little of the time” and “all of the time” (Fig. 5). However, the extent to which the parameters of the six items were similar—expressed as the degree of parallelism of the regression lines—was lower compared with the results of our previous general population sample (N = 32,022; Tomitaka, Kawasaki & Furukawa, 2015b). A possible reason for this difference in parallelism might be the sample sizes. In support of the sample size effect, the lines for the 6 items were highly parallel between “a little of the time” and “some of the time,” but became less so and began fluctuating between “some of the time” and “all of the time,” where the frequencies were much smaller compared to “a little of the time” (Fig. 5).

We assumed that the latent trait of depressive symptoms follows an exponential distribution because this assumption enables us to explain the mathematical patterns of the total scores and item responses. However, the K6 items were developed using item response theory, which generally assumes a normally distributed latent trait (Kessler et al., 2002; Van der Linden & Hambleton, 1997). As noted in the introduction section, there is still little evidence of depressive symptoms being described by a normally distributed latent trait. Claiming that the latent trait of depressive symptom items follows a normal distribution would require a theory explaining how a normally distributed latent trait leads to the observed patterns in this study.

Strengths and limitations

This study has some limitations. First, although we evaluated whether the total item scores and the item responses follow an exponential distribution, we did not consider other possible mathematical models in our analysis. In general, the most important aspect of model evaluation is testing whether a model fits the data better than other models. However, to our knowledge, no other mathematical models have been reported so far. Future studies might seek to evaluate the comparative fit of other models to the data from the National Archive of Data on Aging. Second, participants of the MIDUS did not receive a structured psychiatric interview to diagnose their symptoms. Third, although we found that the total score distribution and item responses of the K6 exhibited exponential patterns, consistent with the screening test for depression (CES-D), K6 is not solely a measure of depression, but a broader measure of psychological distress.

However, our study has a methodological advantage. First, the sample was representative of the general American population, thus ensuring limited selection bias. In addition, four subsamples from different populations were recruited, thus ensuring the reproducibility of the findings. Finally, the present study provides important information on the distribution of depressive symptoms in the general population.

Supplemental Information

Table S1 Item responses of national RDD sample of twin pairs, siblings of individuals from the RDD sample, and oversamples from five metropolitan areas

The item response rates for the remaining three subsamples exhibited a similar pattern as for the national RDD subsample—the highest response frequency was for “none of the time,” after which the response frequency decreased with increasing item score (such that the lowest response frequency was observed for “all of the time.”).

Click here for additional data file.

We would like to thank Professor Kazumasa Mori at Bunkyo University for his helpful advice.

Additional Information and Declarations

Competing Interests

Author Contributions

Data Availability

Shinichiro Tomitaka is an employee of Department of Mental Health, Panasonic Health Center, Tokyo, Japan.

Shinichiro Tomitaka conceived and designed the experiments, performed the experiments, analyzed the data, wrote the paper, prepared figures and/or tables, reviewed drafts of the paper.

Yohei Kawasaki, Kazuki Ide, Maiko Akutagawa, Hiroshi Yamada, Ono Yutaka and Toshiaki A. Furukawa reviewed drafts of the paper.

The following information was supplied regarding data availability:

The raw data and code are freely available to researchers and attributes to the National Archive of Data on Aging: http://midus.wisc.edu/. The data for this study was downloaded from the ICPSR 2760: http://www.icpsr.umich.edu/icpsrweb/NACDA/studies/2760.

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
