# Peer review of "Pattern analysis of total item score and item response of the Kessler Screening Scale for Psychological Distress (K6) in a nationally representative sample of US adults"

_PeerJ, doi:10.7717/peerj.2987_

## Round 0.1 · original submission · Major Revisions

Your article has now been reviewed. Please see the attached reviewers' comments for necessary changes and clarifications.

·

Basic reporting

The paper explores the distribution of K6 scores and items. It is generally clearly written.

1. Line 276-277: “...which uniquely has memoryless property...” – could the authors explain what they mean by this phrase?

2. Line 305: “diagnosis” should be “diagnose”

Experimental design

Analyses of existing data seem appropriate

Validity of the findings

1. An alternative explanation for the findings might be that the terms used in the scales are exponential, rather than the underlying trait being exponential. For example, if the scale used in the K6 was "Not at all", "Almost never", "Hardly ever", "Seldom", "Occasionally", "Some of the time", "All of the time", we would see a different distribution. As the K6 is designed to capture higher risk states rather than healthy states, it may be skewed for capturing higher severity responses (the PHQ-9 is an even better example of this) and have insufficient sensitivity to discriminate at the lower tail of the continuum.

2. It may also be the case that the K6 is not very good at distinguishing milder symptoms, as it does not include symptoms such as irritability, unhappiness, fatigue, problems concentrating. Such symptoms were included in the original item pool used to develop the K10/K6, but excluded as they do not differentiate clinical states as accurately as more severe symptoms. The K6 aims to identify clinical states but the authors are interested in the full latent trait, which may not be the purpose of the K6.

3. The authors focus on depressive symptoms and suggest that the K6 is a measure of depressive symptoms. However, it was designed to be a measure of psychological distress, capturing anxiety in addition to depression.

4. The authors note that IRT “generally” assumes that a normally distributed latent trait underlies the performance of a manifest variable. Is it the case that the assumption of normality is required for all IRT models?

Additional comments

I found it difficult to appreciate the significance of the paper. What are the implications of the findings? Is it just that the assumptions around the modeling of symptoms are flawed because it is not normally distributed? If so, the paper would be strengthened by making a case for an alternative modeling approach. Or are there broader clinical implications for the findings? It is difficult to determine the significance of classifying the distribution of mental health measures – it is already well established that a small proportion of the population experiences depression symptoms and symptoms are severe in a much smaller proportion.

·

Basic reporting

This is a well written and interesting manuscript that adheres to the policies of the journal. However, there are a number of issues that require clarification.
1. In the first paragraph of the introduction the authors say "Because depressive symptoms are closely related to clinical levels of depression...". This sentence is somewhat ambiguous. What do the authors exactly mean? Do they mean that a clinical diagnosis of depression is determined mainly by a count of symptoms? If so, I think the authors could state this more clearly.
2. The main aim of this manuscript is to confirm the exponential distribution of K6 scores. Early on in the introduction the authors simply state that the count of depressive symptoms follows an exponential distribution. However, for the non-statistically oriented readers, this might mean nothing. Perhaps the authors can provide a visual representation of what an exponential distribution looks like. This will make it clearer and more concrete to the reader.
3. Following on from point 2 the authors state that depressive symptoms follow an exponential distribution except at the lowest end of the range of scores. What do the authors mean by this? How is it different at the lowest end of the range of scores?

Experimental design

Given that this manuscript is essentially a statistical exploration of four data sets more detail is required with regards to the statistical analysis. The authors state that on line 150 that "the distribution of the total score were analyzed using a log-normal scale and curve fitting". Could the authors expand on this? When the authors say they "analyzed" this might be more accurately described as "visualized". What curve fiting procedure was used? What do the authors mean by curve fitting using the least squares method? Can the authors point to other examples of what they have done?
In the results section the authors provide parameter estimates for the curve fitting procedure yet there is no explanation for what these estimates mean. The r squared is relatively obvious but what is the y value representative of?

Validity of the findings

The conclusions are appropriate to the findings.

---

## Round 0.2 · Minor Revisions

Minor Revisions required:

* Please refer to Figure 1 in the manuscript.
* The sentence in the first paragraph describing the relationship between clinical depression and depressive symptomology is still unclear. Please consider amending as follows (or similar): "Because depressive symptomology (number and chronicity) form the basis of a diagnosis of clinical depression, there is considerable interest in understanding the distribution of depressive symptoms in the general population."
* Please include the finding about the 'rarely' response at the end of the Results section of the abstract. i.e. "while the rarely response is not related to this exponential pattern."
* Please consider including a brief sentence about the K6 not being solely a measure of depression in the limitations section (and therefore comparisons with the CES-D come with some minor caution). This is in relation to the K6 being a broader measure of psychological distress. You could include evidence that the K6 is a validated screener for a diagnosis of clinical depression using the CIDI.

---

## Round 0.3 · accepted · Accept

Thank you for the final minor revisions.